# Session Rating of Perceived Exertion (sRPE) Load and Training Impulse Are Strongly Correlated to GPS-Derived Measures of External Load in NCAA Division I Women’s Soccer Athletes

**DOI:** 10.3390/jfmk6040090

**Published:** 2021-10-29

**Authors:** Andrew T. Askow, Alexa L. Lobato, Daniel J. Arndts, Will Jennings, Andreas Kreutzer, Jacob L. Erickson, Phil E. Esposito, Jonathan M. Oliver, Carl Foster, Andrew R. Jagim

**Affiliations:** 1Department of Kinesiology and Community Health, University of Illinois at Urbana-Champaign, Urbana, IL 61801, USA; askow2@illinois.edu; 2Department of Kinesiology, Texas Christian University, Fort Worth, TX 76109, USA; a.lobato@tcu.edu (A.L.L.); d.arndts@tcu.edu (D.J.A.); will.jennings28@tcu.edu (W.J.); a.kreutzer@tcu.edu (A.K.); p.esposito@tcu.edu (P.E.E.); 3Sports Medicine, Mayo Clinic Health System, Onalaska, WI 54601, USA; erickson.jacob@mayo.edu; 4Department of Biobehavioral Sciences, Teachers College, Columbia University, New York, NY 10027, USA; jo2667@tc.columbia.edu; 5Department of Exercise and Sport Science, University of Wisconsin-La Crosse, La Crosse, WI 54601, USA; cfoster@uwlax.edu

**Keywords:** athlete monitoring, internal load, ratings of perceived exertion, soccer

## Abstract

Purpose: The purpose of this study was to determine whether session rating of perceived exertion-derived training load (sRPE-TL) correlates with GPS-derived measures of external load in National Collegiate Athletics Association (NCAA) Division I female soccer athletes. Methods: Twenty-one NCAA Division 1 collegiate women’s soccer athletes (11 starters, 10 non-starters; 65.1 ± 7.2 kg, 168.4 ± 7.9 cm, 20.3 ± 1.5 yrs) volunteered to take part in this study. Data for this study were collected over the course of 16 weeks during the 2018 NCAA women’s soccer season. External load and heart rate (HR) data were collected during each training session and match during the season. At least 30 min after the end of an activity (e.g., match or practice), athletes were prompted to complete a questionnaire reporting their perceived exertion for the session. sRPE-TL was calculated at the end of the season by multiplying perceived exertion by the respective session duration. Results: sRPE-TL was very strongly correlated with total distance, distance covered in velocity zones 1–3, the number of accelerations in zones 4 and 5, total PlayerLoad™, and PlayerLoad™. For internal load, sRPE-TL correlated very strongly (0.70 ≤ |r| < 0.90) with Edward’s and Bannister’s TRIMP and strongly (0.50 ≤ |r| < 0.70) with duration spent in in heart rate zones 5 and 6 (80–90% and 90–100% max HR, respectively) while correlations with maximum HR (bpm), mean HR (bpm), and mean HR (%) and sRPE-TL were moderate (0.30 ≤ |r| < 0.50). Conclusions: In NCAA Division I women soccer, sRPE-TL is strongly associated with external measures of workload. These relationships were stronger during match play, with acceleration load and total distance exhibiting the strongest relationship with sRPE-TL.

## 1. Introduction

Progressive overload is critical for maximizing positive adaptations to training. Inadequate overload can cause failure to adapt, while excessive load may increase the risk of injury, illness, non-functional overreaching, or even overtraining syndrome [1]. Thus, load monitoring throughout a season is a crucial component of athlete management. Quantifying training load during sports practice can be challenging, due to the large number of athletes and their unique movement characteristics at any given point throughout the training session or competition. This is particularly true in the setting of team sports, where a given external workload may represent considerable variation in the internal training load, which is likely of more importance as a driver of the adaptive response [2]. A variety of wearable technologies have been developed, of which each offers a unique set of movement kinematics and measures of external workload that can be captured. In particular, wearable global positioning systems (GPS) in combination with inertial sensors have been of particular interest for sport applications due to their ability to quantify kinematic and inertial load metrics in a variety of settings. These systems have a growing body of evidence supporting their use with valid and reliable measures of various parameters of movement kinematics such as velocity, acceleration, and distances covered [3,4,5,6] for a variety of different sports [7]. However, these technologies can be cost-prohibitive and require training and knowledge to efficiently interpret data and reduce error. Further, these units only measure external training load, whereas internal load (i.e., the physiological stress imparted by external load) more likely mediates the adaptive response to training and the strain of training and competition experienced by the athlete [2]. Additionally, each GPS-system may have accompanying software programs that utilize various thresholds or proprietary metrics to quantify speed zones, high-speed distances, and various derivations of training load or player load. As a result, comparisons across systems may prove challenging as the resulting parameters may have been calculated differently. Therefore, other methods of assessing training load may offer additional benefits, particularly if they are calculated using consistent methods, and would therefore be able to be compared universally across different settings.

Historically, heart rate (HR)-based methods have been primarily used for quantifying internal training load. However, these methods require the use of wearable technology in order to record HR. Although HR sensors are much more accessible in terms of cost compared to GPS units, they are still susceptible to technical errors resulting in lost data. Further, HR measures of internal load seem to be fairly inadequate in quantifying load during very high-intensity training [8] and may not adequately account for the range of intensities experienced. Furthermore, they may not account for the internal stressors incurred in collision sports, which consist of high-impact collisions and blocking, tackling, or rushing activities. Thus, solely relying on these measures to estimate cumulative load may result in an inaccurate approximation of load, particularly in a highly stochastic sport such as soccer. In response, Foster and colleagues [9] developed the concept of integrating the Rating of Perceived Exertion (RPE) over the duration of a training session or competition to calculate a session RPE (sRPE), for use in retrospective estimation of global perceived exertion for the entire training bout. When multiplied by the duration of an exercise bout, this duration–intensity product yields an sRPE-derived measure of training load (sRPE-TL) as a subjective measure of internal load. This approach has been shown to provide a low-cost and low-tech method of quantifying internal training load in a variety of sport settings [9,10].

Since its introduction, sRPE-TL has been used in a large number of studies and has been shown to correlate well with other measures of internal load [10,11,12,13,14]. However, most training programs continue to prescribe work in terms of external load. Therefore, it is important to understand the relationship between sRPE-TL and measures of external load by examining its validity and utility as a tool for monitoring training load within a particular team sport setting. Several studies have done so and have provided preliminary evidence to indicate that sRPE-TL is highly correlated with total distance [15], accelerometer load [16], and high-speed efforts [16]. Further, a recent meta-analysis indicates that sRPE-TL is more strongly associated with external load compared to traditional HR-based measures of internal load [17]. However, few studies have assessed these relationships in female athletes competing at the collegiate level. Given the physiological differences between males and females and possible sex differences in the reliability of perceived exertion [18], it is important to further examine the relationship between sRPE-TL and measures of external load in females. Furthermore, it is important to examine these relationships across specific sport types as certain competition demands, movement characteristics, and physical abilities of the athletes may influence the magnitude of the relationship observed. This is particularly true considering the relatively liberal substitution rules within collegiate soccer, which differ meaningfully from the “starters continue to play” FIFA rules seen at the professional level. Therefore, the purpose of this study was to determine whether sRPE-TL correlates with GPS-derived measures of external load in National Collegiate Athletics Association (NCAA) Division I female soccer athletes. As a secondary aim, we sought to determine whether sRPE-TL or HR-based internal load was more strongly correlated to the external load. We hypothesized that sRPE-TL would correlate well with measures of external load.

## 2. Methods

### 2.1. Subjects

Twenty-one NCAA Division 1 collegiate women’s soccer athletes (11 starters, 10 non-starters; 65.1 ± 7.2 kg, 168.4 ± 7.9 cm, 20.3 ± 1.5 yrs.) volunteered to take part in this study. All athletes were screened for health contraindications by the sports medicine staff as part of the team’s normal standard of care and, thus, the sole inclusion criterion was to be a member of the team. Goalkeepers were excluded due to the relatively low total distance traveled compared to other positions. Interested athletes were informed of the risks associated and provided written informed consent prior to participation. All procedures involving human subjects were conducted in accordance with the requirements of the Declaration of Helsinki and approved by the Institutional Review Board at Texas Christian University (approval no. 1807-114-1807; approval date 30 July 2018).

### 2.2. Study Design

The data for this study were collected over the course of 16 weeks during the 2018 NCAA women’s soccer season. External load and HR data were collected during each training session and match during the season. At least 30 min after the end of an activity (e.g., match or practice), athletes were prompted to complete a questionnaire reporting their perceived exertion for the session. sRPE-TL was calculated at the end of the season by multiplying perceived exertion by the respective session duration. On match days, the session durations were reflective of total warm-up duration plus the total (actual) on-field time. Subsequently, the relationships between internal load (sRPE-TL and HR) and external load measures were determined to assess the validity of sRPE-TL. A total of 1767 data points from the GPS units and 1105 perceived exertion questionnaires were collected over the course of the season. However, after excluding goal keepers (*n* = 211), players who left the team early in the season (*n* = 43), and incomplete datasets (i.e., where either sRPE-TL or external load data were not available; *n* = 441), a total of 1072 instances of concurrent sRPE-TL and external load data were identified and included in the analysis. The final analysis included both starters and non-starters.

### 2.3. Preliminary Testing

Two days prior to the beginning of preseason practice, athletes reported to the athletics complex for preliminary testing to determine maximal velocity and HR using methods recommended by Turner and colleagues [19]. For maximal velocity, athletes completed 3 trials of a maximal sprint task with a flying start. Electronic timing gates (Dashr, LLC; Lincoln, NE, USA) were set up 20 m apart and used to calculate maximal velocity. Following the sprint task, athletes completed a Yo-Yo Intermittent Recovery Test Level 1 for the determination of maximal HR, which was later used for the calculation of relative HR variables and HR intensity zones. The test was continued until athletes could no longer complete a 40 m stage in two consecutive attempts. During the test, HR was measured using chest-worn HR sensors (H7; Polar Electro, Kempele, Finland).

### 2.4. Data Collection Procedures

Over the course of the season, the external load was assessed using commercially available GPS units (OptimEye X4; Catapult Innovations, Melbourne, Australia) sampling at 10 Hz. Prior to the first training session, research personnel fitted athletes with a GPS unit in accordance with manufacturer guidelines. The unit was secured between the athlete’s scapulae using a purpose-built cloth garment from the manufacturer. The same GPS unit and garment were used every session for each athlete to minimize the risk of inter-unit variability. Athlete profiles were created within the software program OpenField (Catapult Innovations, Melbourne, Australia), with pertinent demographic data used for each individual and data collected during preliminary testing. Previous research indicated the current GPS monitoring system provides valid and reliable measures of movement kinematic variables [3,5,6]. Specifically, strong relationships (r = 0.99–1.00) with small typical error of the estimate (TEE; 0.08–0.38) values have been reported for total distance, and distances at low speeds (0–3 m∙s^−1^), moderate speeds (3–5 m∙s^−1^), high speeds (5–7 m∙s^−1^), and very high speeds (>7 m∙s^−1^), when compared to criterion measures [5]. There is also evidence indicating a high degree of interunit reliability for peak velocity (ICC = 0.97; Percentage typical error of measurement [%TEM] = 1.6%), and distance covered for low speeds (ICC = 0.97; %TEM = 1.7%) and high speeds (ICC = 0.88; %TEM = 4.8%) [6].

At least 30 min prior to every practice or match, the same research personnel arrived at the athletic complex to prepare the units for data collection and allow for the acquisition of the satellite signal. When athletes arrived and GPS units were connected, external load data were collected using the manufacturer’s data acquisition software. In order to ensure that duration-sensitive data were captured accurately, practice and match activities were coded within the software at the time of collection to minimize the inclusion of rest time surrounding practice or between periods. For matches, only time spent during warmups and time on the field were included in the analysis. Half time and time spent on the sidelines were excluded. Following the conclusion of each session, data were uploaded to a cloud-based version of the software for future analysis. Using the software, PlayerLoad™ was automatically calculated as a proprietary metric used to quantify training load. Accumulated PlayerLoad™ can be calculated using the following formula:$$\mathrm{PlayerLoad}\;{\left(\mathrm{accul}\right)}_{t=n}=$$
$$\overset{t=n}{\underset{t=0}{\sum }}\sqrt{\left[{\left({\mathrm{fwd}}_{t=i+1}{-\mathrm{fwd}}_{t=i}\right)}^{2}+{\left({\mathrm{side}}_{t=i+1}{-\mathrm{side}}_{t=i}\right)}^{2}+{\left({\mathrm{up}}_{t=i+1}{-\mathrm{up}}_{t=i}\right)}^{2}\right]}$$
where *t* is time, fwd is forward acceleration, side is sideways acceleration, and up is upwards acceleration. The metric is a volume-based measure of summated external load using arbitrary units. Similarly, total distance traveled and distance traveling at given velocities (band 1 = 0–6 km∙h^−1^; band 2 = 6–12 km∙h^−1^; band 3 = 12–18 km∙h^−1^; band 4 = 18–25 km∙h^−1^; band 5 = >25 km∙h^−1^) were calculated for the correlational analysis. Finally, acceleration load, the sum of absolute values (i.e., both positive and negative accelerations) for acceleration (a volume measurement of total speed change activity), was calculated using the software as well.

In order to minimize the effect of fatigue immediately following a session, sRPE data were collected at least 30 min following the end of each session using a smartphone app (TeamBuildr; TeamBuildr, LLC, Landover, MD, USA). The sRPE has been shown to be very robust for time post exercise, around a nominal average of 30 min [10,20]. At this point, athletes were visually presented with the scale previously introduced by Foster et al. [9] and reported their RPE for the session. Following the submission of their responses, data were automatically uploaded to cloud-based software. Any responses submitted on the following day were excluded from further analysis. Session duration was then used to calculate sRPE-TL. To further investigate the relationship between sRPE and other measures of load, all session types were stratified by sRPE into high (sRPE ≥ 6) and low (sRPE ≤ 5) difficulty sessions. Specifically, this approach was used to determine if differences in various measures of external workload existed when a threshold of sRPE-TL was used to categorically group sessions into “low” and “high” sRPE-TL.

Using previously established methods [14], additional HR-derived measures of internal training load were calculated. Bannister [21] training impulse (TRIMP) scores were calculated using the following formula:$${\mathrm{TD}\;\times \;\mathrm{HR}}_{R}\;\times \;0{.64\;\times \;e}^{\left(1{.92\;\times \;\mathrm{HR}}_{R}\right)}$$
where TD is the effective training session duration expressed in min and HR_R_ is determined with the following equation:$$\frac{\left({\mathrm{HR}}_{\mathrm{TS}}\;{-\;\mathrm{HR}}_{B}\right)}{\left({\mathrm{HR}}_{\max}\;{-\;\mathrm{HR}}_{B}\right)}$$
where HR_TS_ is the average training session HR and HR_B_ is the HR measured at rest. Additionally, Edwards TRIMP scores were also calculated by assessing the product of the accumulated training duration (expressed in minutes) across the 5 HR zones by a coefficient relative to each zone (Zone 1 = 50–60% of HR_max_; Zone 2 = 60–70% of HR_max_; Zone 3 = 70–80% of HR_max_; Zone 4 = 80–90% HR_max_; Zone 5 = 90–100% of HR_max_) and then summating the final results [22].

### 2.5. Statistical Analyses

Following the conclusion of the season, data from the GPS units and smartphone app were combined for analysis. The relationship between internal and external load measures was assessed using linear regression with 95% confidence intervals. Subsequently, differences in measures of internal and external load, stratified by high and low sRPE sessions, were assessed using a multivariate analysis of variance (MANOVA). Normality was assessed via visual inspection of normal Q-Q plots and skewness/kurtosis values. Homoscedasticity was assessed using Levene’s Test of Equality of Error Variances. Bonferonni post hoc comparisons were calculated when a significant main effect or interaction was identified. All statistical analyses were conducted using the IBM SPSS Statistics for Windows (version 25.0; IBM Corp., Armonk, NY, USA). The strength of correlation coefficients was classified as trivial (|r| < 0.10), weak (0.10 ≤ |r| < 0.30), moderate (0.30 ≤ |r| < 0.50), strong (0.50 ≤ |r| < 0.70), very strong (0.70 ≤ |r| < 0.90), and nearly perfect (r ≥ 0.90) [23].

## 3. Results

Correlation coefficients, *p* values, and 95% confidence intervals for all relationships can be found in Table 1, Table 2, Table 3 below. sRPE-TL was very strongly correlated with total distance, distance covered in velocity zones 1–3, number of accelerations in zones 4 and 5, total PlayerLoad™, and PlayerLoad™. Strong correlations between distance in velocity band 4, high-speed running distance, and accelerations in zones 3 and 6 and sRPE-TL were observed. For internal load, sRPE-TL correlated very strongly (0.70 ≤ |r| < 0.90) with Edward’s and Bannister’s TRIMP and strongly (0.50 ≤ |r| < 0.70) with the duration spent in in heart zones 5 and 6 (80–90% and 90–100% max HR, respectively) while correlations with maximum HR (bpm), mean HR (bpm), and mean HR (%) and sRPE-TL were moderate (0.30 ≤ |r| < 0.50). When considering event type (i.e., matches vs. practices), relationships were generally stronger in matches. sRPE-TL from match data was nearly perfectly correlated with total distance, Edward’s TRIMP, accelerations in velocity zones 4 and 5, and acceleration load, while these relationships were strong for practice. 

When correlations between external load and HR-based measures (i.e., Edward’s and Bannister’s TRIMP) were calculated, results were similar when both matches and practices were considered. Both Edward’s and Bannister’s TRIMP scores were very strongly correlated with distance in velocity zones 1–3, total distance, total PlayerLoad™, number of accelerations in zones 4 and 5, and total acceleration load. However, Edward’s TRIMP was almost perfectly correlated with distance in velocity zone 1, total distance, total PlayerLoad™, number of accelerations in zone 4 and 5, and total acceleration load for matches, while Bannister’s TRIMP score was strongly correlated with these measures.

When sessions were stratified and coded as either high or low perceived exertion, significant differences in duration, total distance, PlayerLoad™, and acceleration load between high and low exertion sessions were observed (see Figure 1A–D). Similarly, sessions with high perceived exertion also resulted in higher absolute and relative mean HR, sRPE-TL, Edward’s TRIMP, and Bannister’s TRIMP values (see Figure 2A–E).

## 4. Discussion

The primary aim of the current study was to examine the relationships between sRPE-TL and select measures of external workload in collegiate women’s soccer. The primary findings indicate that sRPE-TL was strongly associated with both acceleration load and total distance in both matches and practices, which supports our hypothesis. These relationships were stronger during match play than during practice. Additionally, sRPE-TL was strongly associated with the proprietary PlayerLoad™ metric. As a subjective measure of internal load, sRPE-TL offers a low-cost method to monitor workload throughout a season. This aspect of affordability and minimal equipment need has contributed to its popularity as a monitoring tool within sports. The observed relationship between sRPE-TL and measures of the external workload from the current study aligns with those previously reported in soccer [14,16,24,25,26,27] and across a variety of sport types [11,15,28,29,30,31,32,33]. Previous work in semiprofessional soccer athletes indicated that sRPE-TL was strongly associated with measures of external load such as total distance covered and PlayerLoad™ over 44 training sessions [16]. More recently, Marynowicz et al. [24] observed strong associations between sRPE-TL and total distance, PlayerLoad™, and number of accelerations during an 18-week in-season period in youth soccer athletes. However, small-to-moderate relationships were observed between sRPE-TL and measures of match intensity such as high-speed running distance and the number of impacts. Collectively, these findings suggest that sRPE-TL may better reflect total external workload rather than the intensity of work. However, in the current study, sRPE-TL was found to be strongly associated with acceleration load, which could be classified as an intensity-based measure of external workload volume. It is likely these relationships may be variable based on the fitness level of the athlete and level of competition. An important practical consideration when using sRPE-TL is how the duration is being quantified, specifically during match play. Pustina et al. [34] reported differences in sRPE-TL measures based on how the session duration was defined (i.e., including or excluding warm-ups and half-time, only including on-field playing time, etc.) and found that sRPE-TL calculated using only on-field playing time was the best reflection of external workloads incurred during the match play.

An interesting observation from the current study was that when session types were stratified based on ratings of perceived exertion, differences in internal and external measures of workload were evident. We interpret this as indicating that athletes appear to be able to accurately reflect back on their degree of perceived exertion for a given match or training session, whether the workload metric of interest is external or internal. Additionally, this ability to rate the degree of perceived exertion for external and internal measures remained true whether the metric of interest was pertaining to duration or an accumulative-type metric corresponding to the total amount, duration, or volume of work in addition to metrics that were more reflective of the intensity of work (Figure 1 and Figure 2).

A secondary aim of the current study was to determine which measures of internal load (sRPE-TL or HR-based metrics), were more strongly correlated to the external load. The results of the current study indicate that sRPE-TL was more strongly associated with total PlayerLoad™ compared to simple HR measures (i.e., mean or max HR). However, the accumulative HR zone-based measure, Edward’s TRIMP score, appeared to be more strongly associated with select measures of external workload (total distance, PlayerLoad™, kcal, and acceleration load) compared to sRPE-TL (Table 1 and Table 2). While exhibiting a stronger relationship to external workload in the current study, TRIMP scores do require the use of HR-monitoring technology as opposed to the subject measure sRPE-TL. Further, when determining the overall utility of sRPE-TL, it is also important to evaluate its relationship with physiological measures of internal load, as it may provide an alternative indicator of internal stress incurred by the athlete. Recently, Costa et al. [26] observed good convergent validity between sRPE-TL and TRIMP scores throughout 6 weeks of in-season competition in female soccer athletes. Interestingly, the authors also noted distinct differences in %HRpeak values observed across pre-identified sRPE ranges, indicating that sRPE is useful in identifying specific thresholds of exertion across a range of HR intensities during soccer competition. Similarly, Impellizzeri et al. [14] reported strong associations between sRPE-TL and three different HR-based methods of quantifying internal load, which is contradictory to findings from the current study. In the current study, sRPE-TL was more strongly associated with measures of external workload (i.e., PlayerLoad™) than with HR measures, indicating that sRPE-TL may be more sensitive to variations in external workload measures rather than a reflection of HR. However, because of the practical nature of sRPE-TL, it may be a more suitable option for team-sport programs that do not have access to team-based HR monitoring systems. Along these same lines, sRPE-TL may also be easier to use when quantifying accumulated workloads across an entire season, when compared to measures of HR. Because of the relationships between sRPE-TL and measures of both external and internal workload, it may also offer a multipurpose metric of overall load as it is a reflection of both the total work completed and the internal physiological stress incurred.

This study is not without limitations. The current study was conducted in collegiate female athletes competing at the NCAA Division I level and, therefore, it is unknown if similar relationships between sRPE-TL and external workload would exist in female and male athletes competing at different levels. Environmental factors (i.e., ambient temperature and humidity) may influence HR-derived measures; however, these external factors were not accounted for in the current analysis. Similarly, lifestyle stressors, dietary intake, and sleep habits may have also influenced the perceived difficulty of training with a subsequent impact on how measures of external workload correlate with sRPE-TL.

## 5. Conclusions

In NCAA Division I women’s soccer, sRPE-TL is strongly associated with both external and internal measures of workload. These relationships were stronger during match play, with acceleration load and total distance exhibiting the strongest relationship with sRPE-TL. sRPE-TL can serve as a valuable, and perhaps superior, tool for monitoring workload in women soccer athletes when GPS-based or measures of HR are not an option, or to be used in conjunction with one another.

## Figures and Tables

**Figure 1 jfmk-06-00090-f001:**
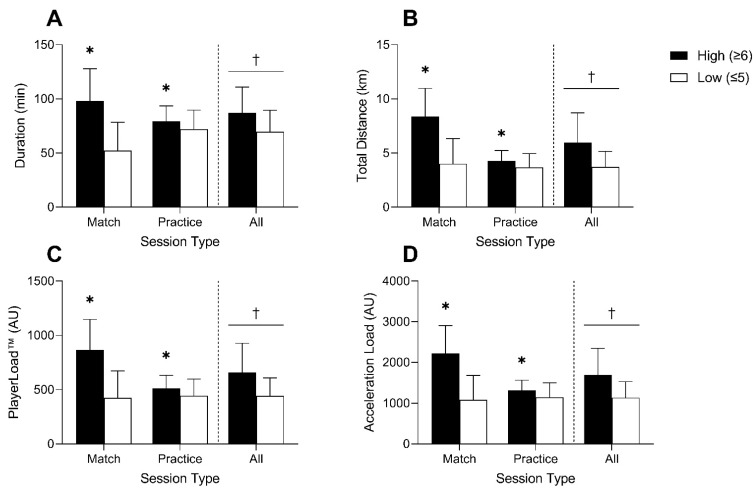
Session duration (**A**), total distance traveled (**B**), PlayerLoad™ (**C**), and acceleration load (**D**) data stratified by high (≥6) or low (≤5) perceived session exertion for matches, practices, and all sessions. * Significant difference between high and low exertion session within session type (*p* < 0.001); † Significant main effect of session exertion (*p* < 0.001).

**Figure 2 jfmk-06-00090-f002:**
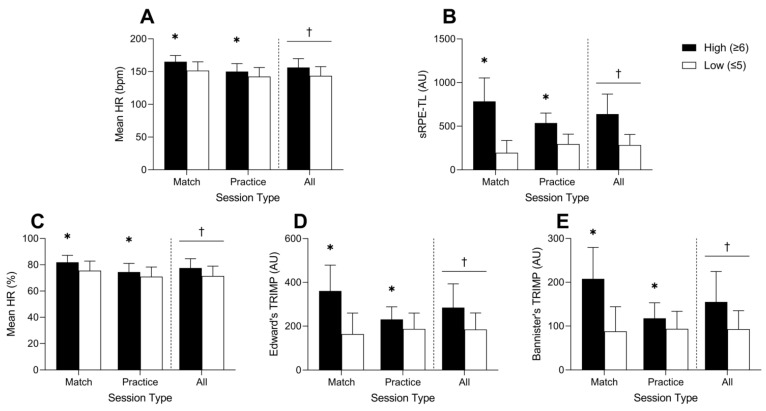
Mean absolute HR (**A**), sRPE-TL (**B**), mean relative HR (**C**), Edward’s TRIMP (**D**), and Bannister’s TRIMP (**E**) data stratified by high (≥6) or low (≤5) perceived session exertion for matches, practices, and all sessions. * Significant difference between high and low exertion session within session type (*p* < 0.001); † Significant main effect of session exertion (*p* < 0.001).

**Table 1 jfmk-06-00090-t001:** Correlations between sRPE-TL and various training load measures for all samples, games, and practices.

	All (*n* = 1072)			Games (*n* = 306)			Practices (*n* = 766)		
Variable	r	95% CI	*p*	r	95% CI	*p*	r	95% CI	*p*
Dur	0.842	0.827, 0.856	<0.001	0.930	0.916, 0.942	<0.001	0.692	0.659, 0.721	<0.001
sRPE	0.840	0.824, 0.854	<0.001	0.785	0.745, 0.818	<0.001	0.876	0.862, 0.889	<0.001
sRPE-TL	N/A	N/A	N/A	N/A	N/A	N/A	N/A	N/A	N/A
Velocity B1 Dist	0.834	0.818, 0.849	<0.001	0.891	0.869, 0.909	<0.001	0.552	0.510, 0.593	<0.001
Velocity B2 Dist	0.814	0.796, 0.830	<0.001	0.877	0.854, 0.897	<0.001	0.520	0.476, 0.562	<0.001
Velocity B3 Dist	0.741	0.717, 0.762	<0.001	0.772	0.731, 0.807	<0.001	0.424	0.374, 0.471	<0.001
Velocity B4 Dist	0.547	0.511, 0.581	<0.001	0.495	0.421, 0.563	<0.001	0.339	0.285, 0.390	<0.001
Velocity B5 Dist	0.257	0.209, 0.303	<0.001	0.146	0.053, 0.237	0.010	0.150	0.091, 0.207	<0.001
Total Dist	0.841	0.826, 0.855	<0.001	0.905	0.887, 0.921	<0.001	0.583	0.542, 0.621	<0.001
Mean Velocity	0.437	0.395, 0.477	<0.001	0.168	0.075, 0.258	0.003	0.124	0.065, 0.182	<0.001
Max Velocity	0.283	0.236, 0.328	<0.001	0.094	0.000, 0.186	0.101	0.248	0.192, 0.303	<0.001
HSR Dist	0.538	0.502, 0.573	<0.001	0.462	0.385, 0.533	<0.001	0.341	0.287, 0.392	<0.001
Meterage per Minute	0.437	0.395, 0.477	<0.001	0.168	0.075, 0.258	0.003	0.124	0.065, 0.182	<0.001
PlayerLoad™	0.827	0.811, 0.843	<0.001	0.892	0.871, 0.910	<0.001	0.551	0.508, 0.591	<0.001
PlayerLoad™∙min^−1^	0.352	0.307, 0.395	<0.001	0.130	0.036, 0.222	0.023	0.096	0.037, 0.155	0.008
kcal Expenditure	0.817	0.799, 0.833	<0.001	0.871	0.846, 0.892	<0.001	0.541	0.497, 0.581	<0.001
kcal∙kg^−1^	0.844	0.828, 0.858	<0.001	0.906	0.888, 0.922	<0.001	0.593	0.553, 0.630	<0.001
HR B1 Dur	−0.061	−0.111, −0.010	0.047	0.067	−0.027, 0.161	0.240	−0.054	−0.113, 0.006	0.136
HR B2 Dur	−0.123	−0.172, −0.073	<0.001	0.083	−0.011, 0.176	0.147	0.041	−0.019, 0.100	0.261
HR B3 Dur	0.035	−0.015, 0.085	0.251	0.335	0.249, 0.416	<0.001	0.296	0.240, 0.349	<0.001
HR B4 Dur	0.386	0.343, 0.428	<0.001	0.495	0.420, 0.563	<0.001	0.370	0.317, 0.420	<0.001
HR B5 Dur	0.690	0.662, 0.715	<0.001	0.693	0.640, 0.739	<0.001	0.462	0.414, 0.508	<0.001
HR B6 Dur	0.567	0.532, 0.600	<0.001	0.485	0.410, 0.554	<0.001	0.342	0.288, 0.393	<0.001
HR B7 Dur	0.152	0.102, 0.201	<0.001	0.162	0.069, 0.252	0.005	0.086	0.027, 0.145	0.017
Max HR (bpm)	0.350	0.305, 0.394	<0.001	0.374	0.290, 0.452	<0.001	0.270	0.213, 0.324	<0.001
Mean HR (bpm)	0.462	0.422, 0.501	<0.001	0.367	0.282, 0.445	<0.001	0.296	0.241, 0.350	<0.001
Max HR (%)	0.281	0.234, 0.327	<0.001	0.255	0.165, 0.341	<0.001	0.224	0.167, 0.280	<0.001
Mean HR (%)	0.425	0.383, 0.466	<0.001	0.311	0.223, 0.393	<0.001	0.269	0.213, 0.323	<0.001
HR Exertion	0.858	0.845, 0.871	<0.001	0.897	0.877, 0.914	<0.001	0.676	0.642, 0.707	<0.001
Exertion Index	0.786	0.766, 0.805	<0.001	0.848	0.820, 0.873	<0.001	0.481	0.434, 0.526	<0.001
Edward’s TRIMP	0.841	0.826, 0.855	<0.001	0.901	0.882, 0.917	<0.001	0.619	0.581, 0.655	<0.001
Bannister’s TRIMP	0.808	0.790, 0.825	<0.001	0.855	0.827, 0.878	<0.001	0.557	0.515, 0.597	<0.001
Accel B1 Count	0.062	0.012, 0.112	0.043	0.049	−0.045, 0.143	0.389	−0.002	−0.061, 0.057	0.956
Accel B2 Count	0.143	0.093, 0.192	<0.001	0.078	−0.016, 0.171	0.174	0.031	−0.029, 0.090	0.395
Accel B3 Count	0.512	0.474, 0.548	<0.001	0.527	0.455, 0.592	<0.001	0.241	0.184, 0.296	<0.001
Accel B4 Count	0.854	0.840, 0.867	<0.001	0.916	0.900, 0.930	<0.001	0.600	0.560, 0.636	<0.001
Accel B5 Count	0.852	0.837, 0.865	<0.001	0.914	0.897, 0.928	<0.001	0.594	0.554, 0.631	<0.001
Accel B6 Count	0.504	0.466, 0.541	<0.001	0.658	0.601, 0.708	<0.001	0.252	0.195, 0.307	<0.001
Accel B7 Count	0.093	0.043, 0.142	0.002	0.094	0.000, 0.186	0.101	0.138	0.079, 0.196	<0.001
Accel Load	0.859	0.845, 0.871	<0.001	0.922	0.906, 0.935	<0.001	0.608	0.569, 0.644	<0.001

Velocity B1 = 0–6 km∙h^−1^, velocity B2 = 6–12 km∙h^−1^, velocity B3 = 12–18 km∙h^−1^, velocity B4 = 18–25 km∙h^−1^, velocity B5 = >25 km∙h^−1^; HR B1 = 0–50%, HR B2 = 50–60%, HR B3 = 60–70%, HR B4 = 70–80%, HR B5 = 80–90%, HR B6 = 90–100%, HR B7 = >100%; Accel B1 = −10–−3 m∙s^−2^, accel B2 = −3–−2 m∙s^−2^, accel B3 = −2–−1 m∙s^−2^, accel B4 = −1–0 m∙s^−^^2^, accel B5 = 0–1 m∙s^−^^2^, accel B6 = 1–2 m∙s^−^^2^, accel B7 = 2–3 m∙s^−^^2^, accel B8 = 3–10 m∙s^−^^2^.

**Table 2 jfmk-06-00090-t002:** Correlations between Edward’s TRIMP and various training load measures for all samples, games, and practices.

	All (*n* = 1072)			Games (*n* = 306)			Practices (*n* = 766)		
Variable	r	95% CI	*p*	r	95% CI	*p*	r	95% CI	*p*
Dur	0.820	0.803, 0.836	<0.001	0.931	0.917, 0.943	<0.001	0.638	0.601, 0.672	<0.001
sRPE	0.608	0.575, 0.638	<0.001	0.603	0.540, 0.660	<0.001	0.447	0.398, 0.494	<0.001
sRPE-TL	0.841	0.826, 0.855	<0.001	0.901	0.882, 0.917	<0.001	0.619	0.581, 0.655	<0.001
Velocity B1 Dist	0.856	0.842, 0.869	<0.001	0.919	0.902, 0.932	<0.001	0.585	0.544, 0.623	<0.001
Velocity B2 Dist	0.849	0.835, 0.863	<0.001	0.882	0.859, 0.901	<0.001	0.647	0.611, 0.680	<0.001
Velocity B3 Dist	0.785	0.765, 0.804	<0.001	0.792	0.754, 0.825	<0.001	0.551	0.508, 0.591	<0.001
Velocity B4 Dist	0.572	0.538, 0.605	<0.001	0.523	0.451, 0.588	<0.001	0.363	0.311, 0.414	<0.001
Velocity B5 Dist	0.277	0.230, 0.323	<0.001	0.167	0.074, 0.257	0.003	0.166	0.107, 0.223	<0.001
Total Dist	0.876	0.863, 0.887	<0.001	0.924	0.909, 0.937	<0.001	0.688	0.656, 0.718	<0.001
Mean Velocity	0.520	0.482, 0.555	<0.001	0.225	0.134, 0.313	<0.001	0.309	0.254, 0.362	<0.001
Max Velocity	0.314	0.268, 0.358	<0.001	0.137	0.044, 0.229	0.016	0.270	0.214, 0.324	<0.001
HSR Dist	0.564	0.529, 0.598	<0.001	0.490	0.415, 0.558	<0.001	0.366	0.313, 0.416	<0.001
Meterage per Minute	0.520	0.482, 0.555	<0.001	0.225	0.134, 0.313	<0.001	0.309	0.254, 0.362	<0.001
PlayerLoad™	0.869	0.856, 0.880	<0.001	0.904	0.886, 0.920	<0.001	0.681	0.647, 0.711	<0.001
PlayerLoad™∙min^−1^	0.453	0.413, 0.493	<0.001	0.176	0.084, 0.266	0.002	0.302	0.247, 0.355	<0.001
kcal Expenditure	0.865	0.852, 0.877	<0.001	0.893	0.873, 0.911	<0.001	0.692	0.660, 0.722	<0.001
kcal∙kg^−1^	0.880	0.868, 0.891	<0.001	0.926	0.911, 0.938	<0.001	0.703	0.671, 0.731	<0.001
HR B1 Dur	−0.335	−0.379, −0.289	<0.001	−0.159	−0.250, −0.066	0.005	−0.472	−0.517, −0.425	<0.001
HR B2 Dur	−0.372	−0.414, −0.328	<0.001	−0.069	−0.162, 0.025	0.227	−0.393	−0.442, −0.342	<0.001
HR B3 Dur	−0.084	−0.134, −0.034	0.006	0.223	0.132, 0.311	<0.001	0.120	0.061, 0.178	<0.001
HR B4 Dur	0.366	0.321, 0.408	<0.001	0.401	0.319, 0.477	<0.001	0.464	0.416, 0.509	<0.001
HR B5 Dur	0.808	0.790, 0.825	<0.001	0.731	0.684, 0.772	<0.001	0.815	0.794, 0.834	<0.001
HR B6 Dur	0.759	0.737, 0.779	<0.001	0.644	0.585, 0.696	<0.001	0.745	0.717, 0.770	<0.001
HR B7 Dur	0.214	0.166, 0.262	<0.001	0.190	0.098, 0.280	<0.001	0.195	0.137, 0.252	<0.001
Max HR (bpm)	0.451	0.410, 0.490	<0.001	0.406	0.324, 0.482	<0.001	0.457	0.409, 0.503	<0.001
Mean HR (bpm)	0.678	0.650, 0.705	<0.001	0.511	0.438, 0.577	<0.001	0.700	0.668, 0.729	<0.001
Max HR (%)	0.457	0.417, 0.496	<0.001	0.347	0.261, 0.427	<0.001	0.534	0.490, 0.575	<0.001
Mean HR (%)	0.696	0.669, 0.721	<0.001	0.503	0.429, 0.570	<0.001	0.761	0.735, 0.785	<0.001
HR Exertion	0.983	0.981, 0.984	<0.001	0.987	0.984, 0.989	<0.001	0.964	0.960, 0.968	<0.001
Exertion Index	0.821	0.804, 0.837	<0.001	0.864	0.838, 0.886	<0.001	0.582	0.541, 0.620	<0.001
Edward’s TRIMP	N/A	N/A	N/A	N/A	N/A	N/A	N/A	N/A	N/A
Bannister’s TRIMP	0.959	0.955, 0.963	<0.001	0.964	0.957, 0.970	<0.001	0.920	0.910, 0.928	<0.001
Accel B1 Count	0.090	0.040, 0.140	0.003	0.090	−0.004, 0.183	0.116	0.011	−0.048, 0.071	0.757
Accel B2 Count	0.132	0.082, 0.181	<0.001	0.052	−0.042, 0.145	0.365	0.030	−0.029, 0.090	0.404
Accel B3 Count	0.532	0.495, 0.567	<0.001	0.532	0.461, 0.596	<0.001	0.283	0.228, 0.337	<0.001
Accel B4 Count	0.862	0.849, 0.874	<0.001	0.916	0.900, 0.930	<0.001	0.632	0.595, 0.666	<0.001
Accel B5 Count	0.868	0.855, 0.880	<0.001	0.918	0.901, 0.931	<0.001	0.652	0.616, 0.685	<0.001
Accel B6 Count	0.526	0.489, 0.562	<0.001	0.675	0.620, 0.723	<0.001	0.288	0.233, 0.342	<0.001
Accel B7 Count	0.093	0.043, 0.143	0.002	0.140	0.047, 0.231	0.014	0.108	0.049, 0.167	0.003
Accel Load	0.887	0.876, 0.897	<0.001	0.928	0.914, 0.940	<0.001	0.706	0.675, 0.735	<0.001

Velocity B1 = 0–6 km∙h^−1^, velocity B2 = 6–12 km∙h^−1^, velocity B3 = 12–18 km∙h^−1^, velocity B4 = 18–25 km∙h^−1^, velocity B5 = >25 km∙h^−1^; HR B1 = 0–50%, HR B2 = 50–60%, HR B3 = 60–70%, HR B4 = 70–80%, HR B5 = 80–90%, HR B6 = 90–100%, HR B7 = >100%; Accel B1 = −10–−3 m∙s^−2^, accel B2 = −3–−2 m∙s^−2^, accel B3 = −2–−1 m∙s^−2^, accel B4 = −1–0 m∙s^−2^, accel B5 = 0–1 m∙s^−2^, accel B6 = 1–2 m∙s^−2^, accel B7 = 2–3 m∙s^−2^, accel B8 = 3–10 m∙s^−2^.

**Table 3 jfmk-06-00090-t003:** Correlations between Bannister’s TRIMP and various training load measures for all samples, games, and practices.

	All (*n* = 1072)			Games (*n* = 306)			Practices (*n* = 766)		
Variable	r	95% CI	*p*	r	95% CI	*p*	r	95% CI	*p*
Dur	0.754	0.732, 0.775	<0.001	0.866	0.840, 0.888	<0.001	0.558	0.515, 0.598	<0.001
sRPE	0.608	0.575, 0.639	<0.001	0.607	0.544, 0.664	<0.001	0.423	0.373, 0.471	<0.001
sRPE-TL	0.808	0.790, 0.825	<0.001	0.855	0.827, 0.878	<0.001	0.557	0.515, 0.597	<0.001
Velocity B1 Dist	0.841	0.826, 0.856	<0.001	0.881	0.858, 0.901	<0.001	0.519	0.474, 0.561	<0.001
Velocity B2 Dist	0.835	0.819, 0.849	<0.001	0.836	0.805, 0.862	<0.001	0.594	0.555, 0.632	<0.001
Velocity B3 Dist	0.794	0.774, 0.812	<0.001	0.783	0.743, 0.817	<0.001	0.509	0.463, 0.552	<0.001
Velocity B4 Dist	0.602	0.569, 0.633	<0.001	0.564	0.496, 0.625	<0.001	0.362	0.310, 0.413	<0.001
Velocity B5 Dist	0.299	0.252, 0.344	<0.001	0.192	0.099, 0.281	<0.001	0.160	0.102, 0.218	<0.001
Total Dist	0.869	0.856, 0.881	<0.001	0.892	0.871, 0.910	<0.001	0.631	0.593, 0.665	<0.001
Mean Velocity	0.579	0.545, 0.611	<0.001	0.308	0.221, 0.391	<0.001	0.315	0.260, 0.367	<0.001
Max Velocity	0.329	0.283, 0.373	<0.001	0.157	0.063, 0.247	0.006	0.271	0.215, 0.326	<0.001
HSR Dist	0.595	0.561, 0.626	<0.001	0.530	0.459, 0.595	<0.001	0.364	0.312, 0.415	<0.001
Meterage per Minute	0.579	0.545, 0.611	<0.001	0.308	0.221, 0.391	<0.001	0.315	0.260, 0.367	<0.001
PlayerLoad™	0.845	0.830, 0.859	<0.001	0.863	0.837, 0.886	<0.001	0.617	0.578, 0.652	<0.001
PlayerLoad™∙min^−1^	0.494	0.455, 0.531	<0.001	0.237	0.146, 0.324	<0.001	0.298	0.243, 0.352	<0.001
kcal Expenditure	0.845	0.830, 0.858	<0.001	0.849	0.820, 0.873	<0.001	0.610	0.572, 0.646	<0.001
kcal∙kg^−1^	0.873	0.860, 0.884	<0.001	0.895	0.874, 0.912	<0.001	0.643	0.607, 0.677	<0.001
HR B1 Dur	−0.254	−0.300, −0.206	<0.001	−0.102	−0.195, −0.008	0.074	−0.354	−0.405, −0.300	<0.001
HR B2 Dur	−0.424	−0.464, −0.382	<0.001	−0.177	−0.267, −0.084	0.002	−0.446	−0.492, −0.397	<0.001
HR B3 Dur	−0.227	−0.274, −0.178	<0.001	0.087	−0.007, 0.180	0.129	−0.062	−0.121, −0.002	0.088
HR B4 Dur	0.283	0.237, 0.329	<0.001	0.314	0.226, 0.396	<0.001	0.400	0.348, 0.448	<0.001
HR B5 Dur	0.756	0.733, 0.777	<0.001	0.662	0.606, 0.712	<0.001	0.742	0.715, 0.768	<0.001
HR B6 Dur	0.792	0.773, 0.810	<0.001	0.693	0.640, 0.739	<0.001	0.756	0.729, 0.780	<0.001
HR B7 Dur	0.293	0.246, 0.338	<0.001	0.313	0.226, 0.396	<0.001	0.287	0.232, 0.341	<0.001
Max HR (bpm)	0.458	0.417, 0.497	<0.001	0.437	0.357, 0.510	<0.001	0.463	0.415, 0.508	<0.001
Mean HR (bpm)	0.737	0.713, 0.759	<0.001	0.613	0.551, 0.669	<0.001	0.766	0.740, 0.790	<0.001
Max HR (%)	0.420	0.378, 0.461	<0.001	0.357	0.272, 0.436	<0.001	0.462	0.414, 0.508	<0.001
Mean HR (%)	0.721	0.696, 0.745	<0.001	0.587	0.521, 0.645	<0.001	0.764	0.738, 0.788	<0.001
HR Exertion	0.955	0.951, 0.960	<0.001	0.966	0.959, 0.971	<0.001	0.903	0.892, 0.914	<0.001
Exertion Index	0.827	0.810, 0.842	<0.001	0.842	0.813, 0.868	<0.001	0.544	0.501, 0.585	<0.001
Edward’s TRIMP	0.959	0.955, 0.963	<0.001	0.964	0.957, 0.970	<0.001	0.920	0.910, 0.928	<0.001
Bannister’s TRIMP	N/A	N/A	N/A	N/A	N/A	N/A	N/A	N/A	N/A
Accel B1 Count	0.108	0.058, 0.157	<0.001	0.117	0.023, 0.209	0.04	0.008	−0.051, 0.068	0.816
Accel B2 Count	0.160	0.110, 0.208	<0.001	0.073	−0.022, 0.166	0.206	0.053	−0.006, 0.112	0.143
Accel B3 Count	0.550	0.514, 0.584	<0.001	0.564	0.497, 0.625	<0.001	0.261	0.205, 0.316	<0.001
Accel B4 Count	0.822	0.805, 0.837	<0.001	0.852	0.824, 0.876	<0.001	0.554	0.512, 0.594	<0.001
Accel B5 Count	0.833	0.817, 0.848	<0.001	0.861	0.835, 0.884	<0.001	0.581	0.540, 0.619	<0.001
Accel B6 Count	0.513	0.475, 0.549	<0.001	0.667	0.611, 0.716	<0.001	0.257	0.201, 0.312	<0.001
Accel B7 Count	0.092	0.042, 0.142	0.003	0.167	0.074, 0.257	0.003	0.099	0.040, 0.158	0.006
Accel Load	0.861	0.847, 0.873	<0.001	0.883	0.861, 0.903	<0.001	0.626	0.588, 0.661	<0.001

Velocity B1 = 0–6 km∙h^−1^, velocity B2 = 6–12 km∙h^−1^, velocity B3 = 12–18 km∙h^−1^, velocity B4 = 18–25 km∙h^−1^, velocity B5 = >25 km∙h^−1^; HR B1 = 0–50%, HR B2 = 50–60%, HR B3 = 60–70%, HR B4 = 70–80%, HR B5 = 80–90%, HR B6 = 90–100%, HR B7 = >100%; Accel B1 = −10–−3 m∙s^−2^, accel B2 = −3–−2 m∙s^−2^, accel B3 = −2–−1 m∙s^−2^, accel B4 = −1–0 m∙s^−2^, accel B5 = 0–1 m∙s^−2^, accel B6 = 1–2 m∙s^−2^, accel B7 = 2–3 m∙s^−2^, accel B8 = 3–10 m∙s^−2^.

## Data Availability

The data presented in this study are available on request from the corresponding author.

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
