# Peer review of "Session Rating of Perceived Exertion (sRPE) Load and Training Impulse Are Strongly Correlated to GPS-Derived Measures of External Load in NCAA Division I Women’s Soccer Athletes"

_jfmk, 2021, doi:10.3390/jfmk6040090_

Round 1

Reviewer 1 Report

Overall, a concise and well-articulated manuscript that provides practically relevant information to collegiate soccer coaches. 

Comments:

Define sRPE in the title prior to acronym

Define sRPE-TL in the abstract prior to acronym.

Page 3, line 101: Given the time delay between data collection and this manuscript, please indicate whether IRB approval was prospective or retrospective.

Page 3, line 105: I think it is important that some additional information on the team is provided to improve the interpretation of the findings. Please include the teams ranking (e.g. RPI), conference/final season record and any other pertinent information. Did they win conference/make it to the NCAA tournament?

Page 3, line 134: Please provide information on the GPS units. What was the sampling frequency/rate of the unit and the built in accelerometer and is this valid/reliable for quantification of acceleration/deceleration and constant velocity running? How long were the GPS units on before beginning the game to acquire signal? This is important as there could be differences in the relationships between internal and quantified external load depending on the sampling frequency of the GPS units. 

Page 4, Line 148: Was there a cut-off for minutes played in matches to be included in the final analyses? Perhaps provide a range of minutes played for the 1072 data points assessed. 

Page 4, line 150: Please provide the equation for player load. Others have used player load to assess external load (e.g. Wells et al. 2015, PMID 26288394). It would be handy to know whether the calculations are the same for comparative purposes in collegiate women.

Page 4, line 154: This sentence is a little unclear. It reads as if you summed total distance with total distance in all velocity zones. I suspect this is not the case. Please clarify. Also, have you considered summing distance covered in the velocity bands to low-intensity running and high-intensity running for correlation analyses. Other authors have done this and I think it adds to the analyses?

Page 4, line 187: Please explain the rationale for the low/high split on sRPE? How was this determined? If someone rated the session as a 5 vs. 6, practically speaking what is the potential magnitude of difference between these two internal loads?

Please check the variable “HR B7 Du” for correctness in all tables

Please check the values for the 95% CI’s in the tables. This information does not look complete.

Please provide either p-values of p-value ranges (e.g. * =  p<.05, ** =  p<.01, *** = p<.001) for variables in figures 1 and 2.

Author Response

Thank you for taking the time to review our manuscript. Please see the attached document for responses to your comments.

Reviewer 2 Report

The present study is of interest to investigate whether sRPE-TL correlates with GPS-derived measures of external load in National Collegiate Athletics Association (NCAA) Division I female soccer athletes. As a secondary aim, was to determine whether sRPE-TL or HR-based internal load was more strongly correlated to external load. 

As the main conclusion of this work, the authors found that In NCAA Division I women soccer, sRPE-TL is strongly associated with external measures of workload.

Despite the interesting work and upon sound study design, especially in female athletes, I strongly suggest following the comments to improve the quality of the manuscript.

The manuscript is generally well written.

Methods

  1. Please include the reliability and validity results and respective reference(s) of all metrics provided by the Catapult system.
  2. Line 186-187. "To further investigate the relationship between sRPE and other measures of load, all session types were stratified 186 by sRPE into high (sRPE ≥ 6) and low (sRPE ≤ 5) difficulty sessions." Please be more specific and explain why these "cohort values" were made. 
  3. All variables presented normal distribution? 

Results

  1. Authors should present all descriptive results (i.e, mean, SD, median IQR) for each variable. 

Discussion

  1. The authors should present all limitations and practical applications of this study. Remember, that it is important to understand how this study may improve and help technical and medical staff and the athlete during all sports seasons/environments. 
  2. In fact, could be this study generalized for all female soccer athletes?  

Author Response

(The authors gave the same response as above.)

Reviewer 3 Report

The aim of this manuscript is twofold: i) establish potential relationship between training load based on sRPE and GPS-dervied metrics in both games and practices; ii) demonstrate whether sRPE training load and HR-based intrnal load was correlated with external load based on GPS-derived metrics. 

This manuscript appears interesting and provides some field-based information for soccer practitioners. Moreover, it is well written with a logical and clear structure within the introduction and discussion sections. However, it can be improved with some specific changes throughout the text.

Some specific suggestions are provided below:

Introduction

This section is well-structured, but I feel that more details about GPS-related use would better reinforce the study rationale. 

Please consider the following papers to address it:

-GPS data reflect players' internal load in soccer

doi: 10.1109/ICDMW.2017.122

-Characterization of In-season Elite Football Trainings by GPS Features: The Identity Card of a Short-Term Football Training Cycle

doi: 10.1109/ICDMW.2017.122

Statistical Analyses

Did the authors check for all MANOVA's assumptions? Please provide a specific statement. 

Discussion

I suggest the authors to also discuss whether the hypothesis was verified. 

Please, also report potential study limitations at the end of the discussion section.

Good luck!

Author Response

(The authors gave the same response as above.)

Round 2

Reviewer 1 Report

Authors have adequately responded to my comments.